# Visibility Parameter in Sand/Dust Storms' Radio Wave Attenuation Equations: An Approach for Reliable Visibility Estimation Based on Existing Empirical Equations to Minimize Potential Biases in Calculations

**Hamzah N. Mahmood** and **Widad Ismail** *

Auto ID Laboratory, School of Electrical and Electronic Engineering, Universiti Sains Malaysia, Nibong Tebal 14300, Pulau Pinang, Malaysia; hamzahnalrawi@gmail.com
* Correspondence: eewidad@usm.my

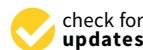

**Featured Application:** Enhancing the techniques of existing empirical equations by relating visibility to Sand/Dust mass concentration for reliable estimations and to minimize potential biases in incorporating calculations.

**Abstract:** In efforts to structure an expression for wave attenuation under a sand/dust storm, most established calculations pronounce optical visibility as an essential parameter. Although visibility information can be retrieved from weather stations, other commonly encountered sources may present it differently, i.e., as total suspended particles (TSP). Consequently, several empirical equations linking visibility to TSP concentrations were evaluated to address offset tendencies in estimations. In addition to substantiating specific equations, the results revealed that averaging a pair of equations has a 46.09% chance of estimating visibilities with a probability of 37.27%, a relatively low error compared to that achieved by employing single equations, which were found to have a probability of 28.93% with a lesser chance (29.58%) of a low estimation error for the same set of data. The resulting enhancement was evaluated by considering a study on a wireless sensor network's (WSN's) signal performance under vaguely labelled meteorological conditions. The meteorological conditions were converted to visibility using the results' suggestions and were found to be in good agreement with an observation standard set by the China Meteorological Administration (CMA) for sand/dust storm outbreak classifications.

**Keywords:** sand/dust storm (SDS); signal attenuation; visibility; total suspended particles (TSP)

## 1. Introduction

The effect of sand storms, among other phenomena, on signal propagation has gathered notable attention from many studies as communication and distant sensing are being utilized by many civil applications. As sand storms are a common meteorological phenomenon that will interfere with the highly anticipated advantages of wireless connectivity, different authors have attempted to structure mathematical representations of path attenuation to estimate the loss of signal energy, particularly when under unique environments.

Chu [1] developed the first known attenuation coefficient (*A*) expression for propagating waves in dust or sand storms by employing related scattering theories and assuming a monodispersive particle

size distribution. Following a series of assumptions, derivations, and substitutions, the author arrived at a general model containing the following parameters:

$$A = 2 * \frac{2\pi}{\lambda} \alpha_0 * r * \Im\left(\frac{\varepsilon - 1}{\varepsilon + 2}\right) \text{dB/km,} \tag{1}$$

where $\lambda$ is the wavelength in meters ($2\pi/\lambda$ = free space wavenumber), $\alpha_0$ is the optical attenuation coefficient (dB per km), $r$ is the particle radius in meters, and $\Im$ of the parenthesized expression corresponds to the imaginary part of the refractive index of the volume of space containing particulates; and $\varepsilon$ is the complex relative permittivity of the particles given by $\varepsilon' - j\varepsilon''$.

Furthermore, Chu [1] demonstrated mathematically, before obtaining the attenuation coefficient expression, $\alpha_0$'s relationship to total particle density ($N_T$), and described it as follows:

$$N_T = 3.67 * 10^{-5} \frac{\alpha_0}{r^2} \text{ (\# of particles/m}^3\text{),} \tag{2}$$

The previous relationship helps solve for $\alpha_0$, thus Chu's [1] attenuation coefficient expression may alternately be expressed as:

$$A = 3.43 * 10^5 \frac{N_T}{\lambda} * r^3 * \Im\left(\frac{\varepsilon - 1}{\varepsilon + 2}\right) \text{dB/km,} \tag{3}$$

However, $\alpha_0$ and $N_T$ in (2) and (3) impose a challenge in terms of fulfilment either due to difficulties involved with their measurements or because they are statistically very scarce [2]. Therefore, it has been suggested that visibility should be considered as an alternative parameter as mathematical expressions relate it to $\alpha_0$ and $N_T$. Chu [1] gave the simple relation between the optical attenuation coefficient ($\alpha_0$) and visibility ($V$) in km as:

$$\alpha_0 = \frac{15}{V} \text{ dB/km,} \tag{4}$$

where the proportionality constant of 15 dB is simply the 10-log of 0.031, which is a measured median of the normalized difference in luminance between a mark and a reference background [1]. Patterson and Gillette [3], on the other hand, developed an empirical relationship between the dust mass concentration ($M$) in g/m$^3$ and the visibility ($V$) in km:

$$MV^\gamma = C, \tag{5}$$

where $\gamma$ and $C$ (g m$^{-3}$ km) are constants, and according to Patterson and Gillette [3], their value depends on the dust storm's geographical area of origin and associated climatic conditions. Essentially, Equation (5) is the ratio of the mass concentration, $M$, to the extinction due to particles, $\sigma_P$, which is designated by $K$. At the same time, the ratio can be related to Koschmieder's visibility law ($V = |\ln 0.02|/\sigma$) by equating $\sigma_P$ to the total extinction $\sigma$–$\sigma$ that includes gaseous as well as particulate optical effects, which is valid for low visibility conditions ($\sigma_p \gg$ extinction by the air molecules; $\sigma_p = \sigma$) [4]. As a result, we obtain:

$$V = \frac{3.912 \, K}{M} = \frac{C}{M} \text{ km,} \tag{6}$$

where $3.912K$ is denoted by $C$, leading to the relation $MV = C$. Assuming a consistent particle size distribution changes with increasing or decreasing visibility, Patterson and Gillette [3] empirically fitted the measurement data with $\gamma$, thus relating $M$ and $V$ as demonstrated in Equation (5).

The relationship given by Equation (5) is often used to solve for $N_T$, assuming a case of polydispersive particle distribution. To illustrate, the particle size distribution term [5] is introduced into Equation (3), which becomes:

$$A = \frac{8.19 * 10^4}{\lambda} * \Im\left(\frac{\varepsilon - 1}{\varepsilon + 2}\right) * \int_0^\infty n(r) \, r^3 dr \, dB/km, \tag{7}$$

where $n(r) \, dr$ is the number of sand/dust particles in the radii range of $r \to r + dr$. A cubic meter of all dust particles per cubic meter of air, i.e., the total relative volume ($v_r$), is given by

$$v_r = \frac{4}{3}\pi \int_0^\infty r^3 n(r) dr \, \text{m}^3 \text{ of dust/m}^3 \text{ of air.} \tag{8}$$

At the same time, $v_r$ has the following relation to the dust mass concentration ($M$) and the mass particle density ($\rho$) in g/m$^3$

$$v_r = \frac{M}{\rho} \, \text{m}^3 \text{ of dust/m}^3 \text{ of air} \tag{9}$$

Using Equations (8) and (9), $v_r$ may alternately be expressed as

$$v_r = \frac{M}{\rho} = \frac{C}{\rho \, V^\gamma} \, \text{m}^3 \text{ of dust/m}^3 \text{ of air.} \tag{10}$$

Considering the form of Equation (8) and the relation given in Equation (10) for the size distribution in Equation (7), we get

$$A = \frac{8.19 * 10^4}{\lambda} * \Im\left(\frac{\varepsilon - 1}{\varepsilon + 2}\right) * \frac{C}{\rho V^\gamma} \, dB/km, \tag{11}$$

Taking advantage of the established relationships and the equations mentioned above, the various authors that have attempted modifying Chu's [1] attenuation model adopted the visibility parameter, and below are some of the known attenuation models constructed by them:

1.  Dong et al.'s [6] attenuation model:

$$A = 2.573 * 10^{-3} \frac{F}{V^\gamma} * \Im\left(\frac{\varepsilon - 1}{\varepsilon + 2}\right) \, dB/km \tag{12}$$

2.  Goldhirsh's [7] attenuation model:

$$A = 2.317 * 10^{-3} \frac{1}{\lambda V^\gamma} * \Im\left(\frac{\varepsilon - 1}{\varepsilon + 2}\right) \, dB/km, \tag{13}$$

3.  Ahmed et al.'s [2] attenuation model:

$$A = 5.670 * 10^4 \frac{1}{\lambda V} * r_e * \Im\left(\frac{\varepsilon - 1}{\varepsilon + 2}\right) \, dB/km, \tag{14}$$

4.  Ahmed's [8] attenuation model:

$$A = 0.629 * 10^3 \frac{F}{V} * r_e * \Im\left(\frac{\varepsilon - 1}{\varepsilon + 2}\right) \, dB/km, \tag{15}$$

5.  Alhaider's [9] attenuation model:

$$A = 0.189 * \frac{F}{V} * r * \Im\left(\frac{\varepsilon - 1}{\varepsilon + 2}\right) \, dB/km, \tag{16}$$

6.　　Dong et al.'s [10] attenuation model:

$$A = 7.7 * 10^{-3} \frac{F}{V^{\gamma}} * \Im\left(\frac{\varepsilon - 1}{\varepsilon + 2}\right) \text{dB/km,} \tag{17}$$

where $F$ is the frequency in GHz in (12), (15), (16), and (17), $V$ is the visibility in km in (12) to (17), $r_e$ is the effective particle radius in µm in (14) and (15), $r$ is the particle radius in µm in (16), and $\lambda$ is the wavelength in meters in (13) and (14).

It is evidently clear that the attempts are very persistent in designating visibility as a critical factor in signal attenuation calculations and as indicative of sand/dust storms intensity. However, while statistical information on dust storm visibility can be retrieved from weather stations, other commonly encountered sources may offer visibility in different terms. For instance, Patterson and Gillette's [3] work, as briefly hinted at earlier, attempted to deduce visibility from dust concentration in the air based on an empirical relationship.

In fact, several additional empirical relationships have been established (Table 1) to calculate visibility ($V$) in km from a known dust concentration ($M$; occasionally, referred to as total suspended particles ($TSP$) in µg/m$^3$. However, there are significant accompanying uncertainties, one of which is the tendency to over- or under-estimate visibility [11], which is this paper's interest.

**Table 1.** Visibility empirical equations, relationship development highlights, and their in-paper references.

| Author(s) | Visibility, km | In-Paper Ref. | Relationship Development Highlights |
|---|---|---|---|
| Chepil and Woodruff [12] | $V = \frac{7078}{M^{\frac{4}{5}}}$ | E1 | • Technique employed: field experiments.<br>• Monitoring period: 2 years.<br>• Location: near Menno, Kansas, USA.<br>• Field Observations: 22 averaged measurements at different heights.<br>• Distance from targeted dust source: at, or very close to, eroding dust source. |
| Patterson and Gillette [3] | $V = \frac{10507}{M^{\frac{100}{107}}}$ | E2 | • Technique employed: field experiments exploring soil-derived aerosols.<br>• Monitoring period: 2 years.<br>• Location: rural areas of west Texas, USA.<br>• Field Observations: 13 measurements at a fixed height.<br>• Distance from targeted dust source: at, or very close to, eroding source.Data best fit: $\frac{100}{107}$ (min: $\frac{25}{29}$, max: $\frac{50}{49}$) |
| Tews [13] | $V = \frac{2032}{M^{\frac{1000}{877}}}$ | E3 | • Information inaccessible. |
| Shao et al. [14] | $V_{<3.5\ km} = \frac{4394515.462}{240.41 M^{\frac{25}{21}}}$ $V_{\geq 3.5\ km} = \frac{-100 \ln(M) + 762}{11}$ | E4 | • Technique employed: datasets from weather stations.<br>• Monitoring period: March/April 2002.<br>• Location: China, Japan, and Korea.<br>• Field Observations: Synoptic records and visibility observed at around 1200 weather stations + near-surface dust concentrations measured at 12 monitoring sites.<br>• 2 likely empirical relationships were found, with respective $R^2$ being 0.68 (V < 3.5 km) and 0.72 (V ≥ 3.5 km). |
| Baddock et al. [15] | $V = \frac{3553.405}{M^{\frac{125}{127}}}$ | E5 | • Technique employed: datasets from weather stations.<br>• Monitoring period: records of 24 years.<br>• Location: rural Mildura/Buronga.<br>• Distance from targeted dust source: 10–100 km from the sandy soils of the Mallee region.<br>• An empirical relationship with an $R^2$ = 0.79. |

With the various assumptions made during attenuation expression derivations, it is necessary to minimize biases that may arise from the encapsulated parameters due to their associated uncertainties. Variations in $M$ will result in different values for $V$ and, consequently, for attenuation ($A$) in Equation (3) via Equations (2) and (4), or in Equation (11) via Equations (5) and (7)–(10).

For convenience, the equations in Table 1 will be referred in discussions and some graphs in this paper as denoted under the third column, where each equation is assigned an alphanumeric character (e.g., E1 corresponds to Chepil and Woodruff's equation).

## 2. Approach and Associated Methods

In this section, the first sub-section presents an overview of datasets gathered from different regions containing measured TSP concentrations and their corresponding recorded visibilities. In addition, the TSP values are substituted into the empirical equations in Table 1 to illustratively present variations in estimations.

Consequently, the next sub-section elaborates on the method, with the help of measured TSP concentrations, with the aim of finding the closet visibility value to that recorded initially based on equations in Table 1.

### 2.1. TSP and Visibility Datasets

Several sets of data point measurements have been collected from different sources (Table 2). Diversifying collected datasets provides region-independent estimations and the viability of drawing generalized conclusions. It is worth noting that the values tabulated represent the minimums and maximums of the collected data. The data points between the extrema are generally known to obey the form of a power function [16].

**Table 2.** Measured total suspended particles (TSPs) concentration ranges and their corresponding recorded visibilities from different sources/regions.

| Area of Sampling | Measured TSP Concentration Range, µg/m³ | Recorded Visibility Range, km | Number of Data Points | Site Type | Reported Measurement Errors | |
|---|---|---|---|---|---|---|
| | | | | | TSP, ug/m³ | Visibility, km |
| KSA [17,18] | 269.86–7639.09 | 0.2–10 | 16 [a] | Urban | ±0.1 | ±10% [c] |
| USA [3,19] | 50–395 | 4.83–56.31 | 14 | Urban | Min [b]: $848.18V^{-0.943}$ Max [b]: $1218.6V^{-0.661}$ | Not specified |
| | 2000–440,000 | 0.06–8 | 13 | Rural | Min: $10507V^{-1.16}$ Max: $10507V^{-0.98}$ | Not specified |
| North-East Asia [13,20] | 43.95–1401.5 | 1.41–12.41 | 23 | Rural | Not specified | ±10% [c] |
| | 101.5–3071.4 | 0.97–24.17 | 30 [a] | Various sites | Not specified | ±10% [c] |
| Australia [15] | 24.99–20,505.5 | 0.2–40.31 | 47 [a] | Rural | Not specified | ±10% [c] |

[a] data point measurements were extracted from images using the digitization [21] method (TSP: Figure 7 in [17]/Figure 9 in [18] and visibility: Figure 8 in [18]; Figure 2 in [14]; Figure 5 in [15]). [b] minimum ($R^2 = 0.7321$) and maximum ($R^2 = 0.7634$) errors are fitted with a trendline for conciseness. Raw data may be inspected individually in its respective reference. [c] an assessment [22] found that modern visibility sensors, despite their accuracy claims, usually operate within an accuracy of ±10%.

The concisely listed variations in Table 2 confirm the notion of difficulty in producing a general empirical relationship. This is observed in Figure 1, which is a visualization of quantities in Table 2, where, for example, Chung et al.'s [20] and Almuhanna's [17,18] rather similar visibilities correspond to two different TSP concentrations in the air. However, despite the noted variations, the data plotted in Figure 1 confirms an inversely-proportional relationship between the visibilities and TSP concentrations, which generally agrees with the relationship of visibility (V) and TSP concentration (*M*) parameters in the equations listed in Table 1.

Accounting for possible errors in measurements, Table 2 is supplied with additional information in that regard. However, despite the lack of thorough documentation for some studies, it is believed that most of the sources cited in this work hold great credibility, particularly the recent ones, given the different techniques employed to conduct robust measurements [23].

### 2.2. Addressing Offsets in Empirical Equation Results

By substituting the TSP concentrations retrieved from the datasets into the equations in Table 1, it becomes evident that the resulting visibilities (Figure 2) generally agree with the recorded values in the distribution form, but suffer from considerable offsets. Furthermore, performing RMSE analysis (Table 3) revealed that there was a significant performance difference among the evaluated models.

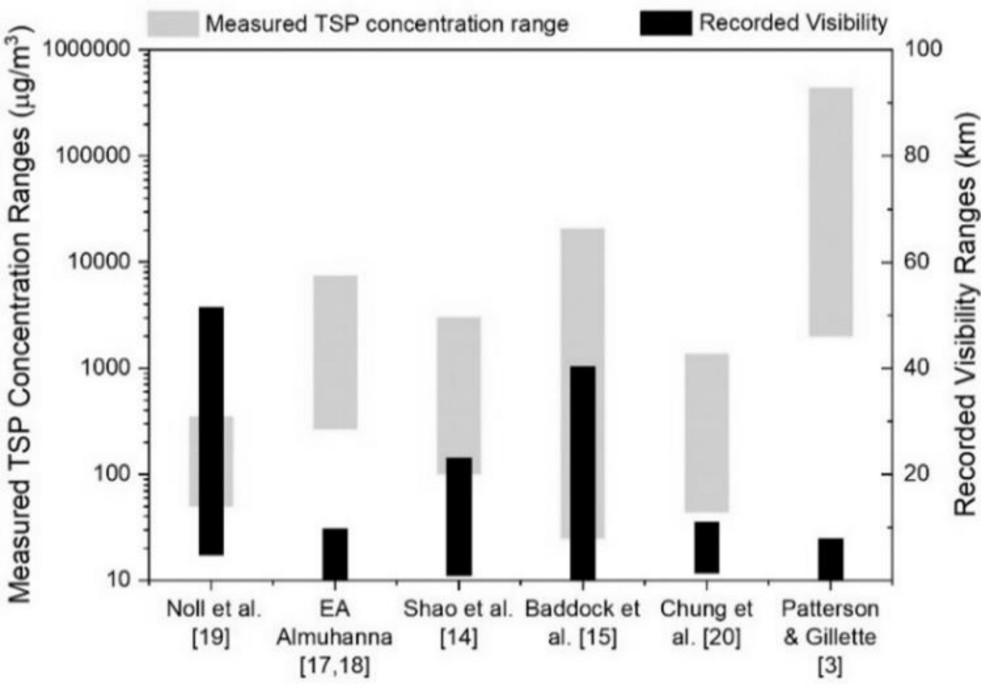

**Figure 1.** Recorded visibility (linear scale) vs. recorded TSP concentration (logarithmic scale) ranges; data point measurement ranges from 6 different sources.

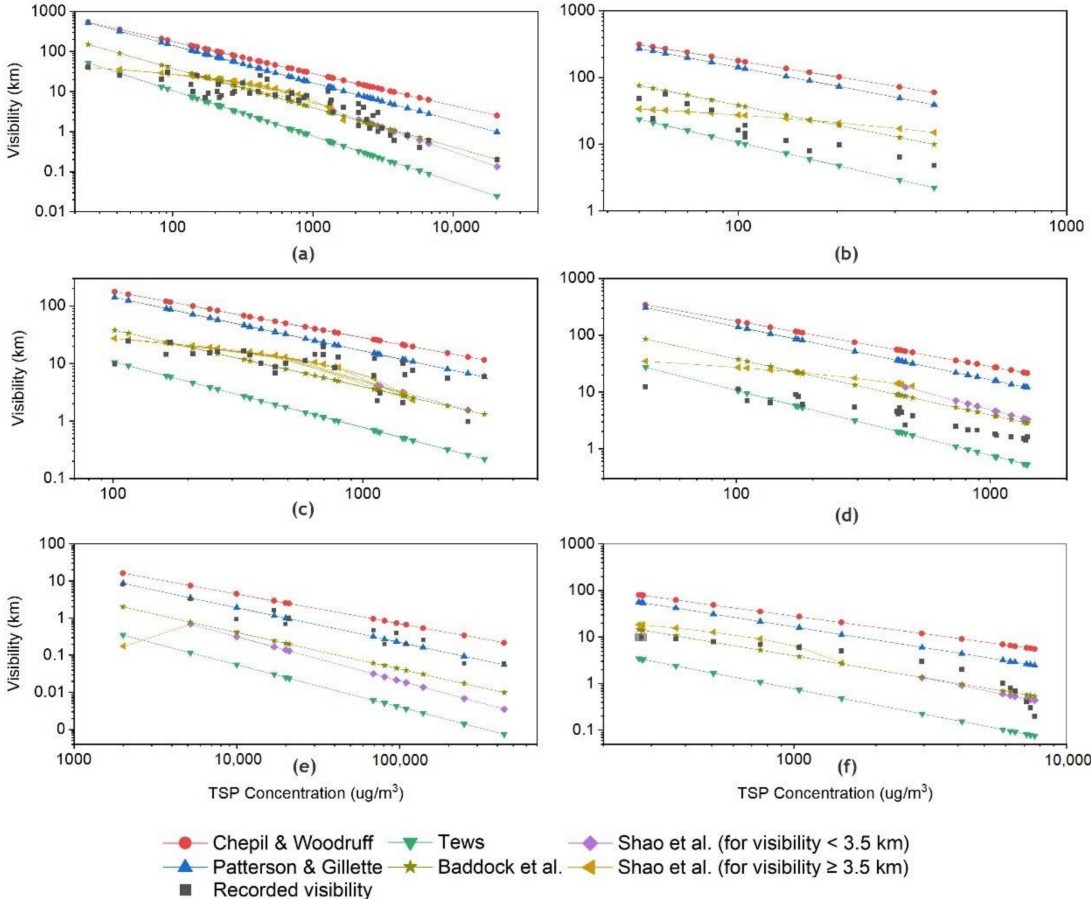

**Figure 2.** Logarithmically plotted results of listed equations (refer to legend) in Table 1 and TSP concentration from Australia (**a**), USA (**b**) and (**e**), North-East Asia (**c**) and (**d**), and KSA (**f**). It can be observed that empirical formulas over- or under-estimate the originally recorded visibilities (▪).

**Table 3.** Performance (RMSE) of the equations in Table 1 compared against recorded visibilities using the TSP values from sources in Table 2.

| Source | E1 | E2 | E5 | E3 | E4 |
|---|---|---|---|---|---|
| KSA [17,18] | 35.93 | 22.55 | 2.10 | 4.42 | 4.18 |
| USA [3,19] | 167.51 | 134.92 | 20.07 | 15.04 | 12.69 |
| | 2.87 | 0.38 | 1.87 | 2.37 | 2.35 |
| North-East Asia [14,20] | 101.54 | 82.02 | 18.81 | 3.58 | 11.14 |
| | 58.26 | 40.66 | 7.40 | 9.81 | 5.71 |
| Australia [15] | 107.41 | 93.61 | 19.94 | 7.72 | 6.35 |
| Average | 78.92 | 62.36 | 11.70 | 7.16 | 7.07 |

However, the variation in performances in the form demonstrated in Figure 2 implies the tendency to either over- or under-estimate visibilities based on given TSP concentrations. Therefore, it indicates a chance that the closest values may reside within the average of two equations.

To develop an algorithm addressing the noted offsets in the calculated visibilities, we first defined a layout for the retrieved data of calculated and recorded visibilities. By breaking the layout into specific segments, the data became hospitable for transitioning into a programming environment. Figure 3 highlights the breakdowns, with brief descriptions, of the tabulated data assigned to the arrayed variables.

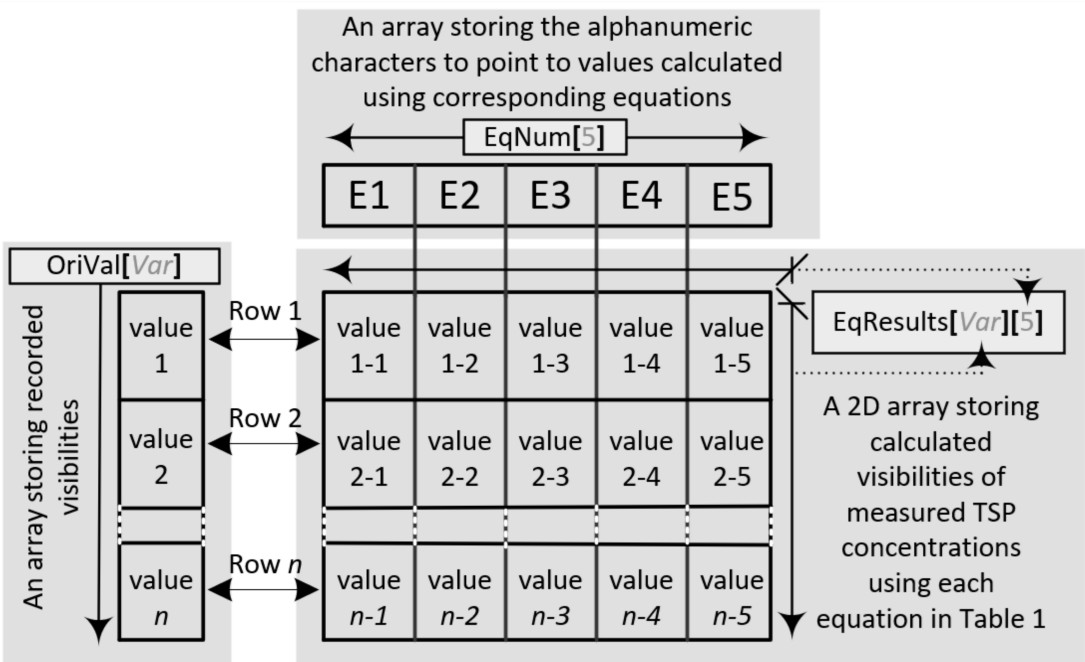

**Figure 3.** Breakdown of tabulated data for coding interpretability.

Three arrays were defined to hold the necessary data: namely, OriVal[*Var*] to store the recorded visibilities, EqResults[*Var*] [5] to store the results of the equations in Table 1, and EqNum [5] to store the alphanumeric characters to track alphanumerically the final, 2-averaged equations, with the lowest error percentage. Additionally, the *Var* variable was substituted by the number of rows until Row *n*, thus setting a boundary for loops and associated counters.

As demonstrated in Figure 4, the program comprises a few sequential steps, starting with initializing the necessary variables and data-holding arrays. The program, then, processes one row at a time (i.e., Row 1 to Row *n* in Figure 3), where each row has two main parts; the first is the original value (e.g., Value 1 in Figure 3) of the recorded visibility, and the second part is a corresponding row of

calculated values (Value 1-1 to Value 1-5) according to each equation (E1 to E5). The steps are repeated for the subsequent rows, and stored values are retrieved from the arrays via the included counters (*colcounter* and *resultsCol*).

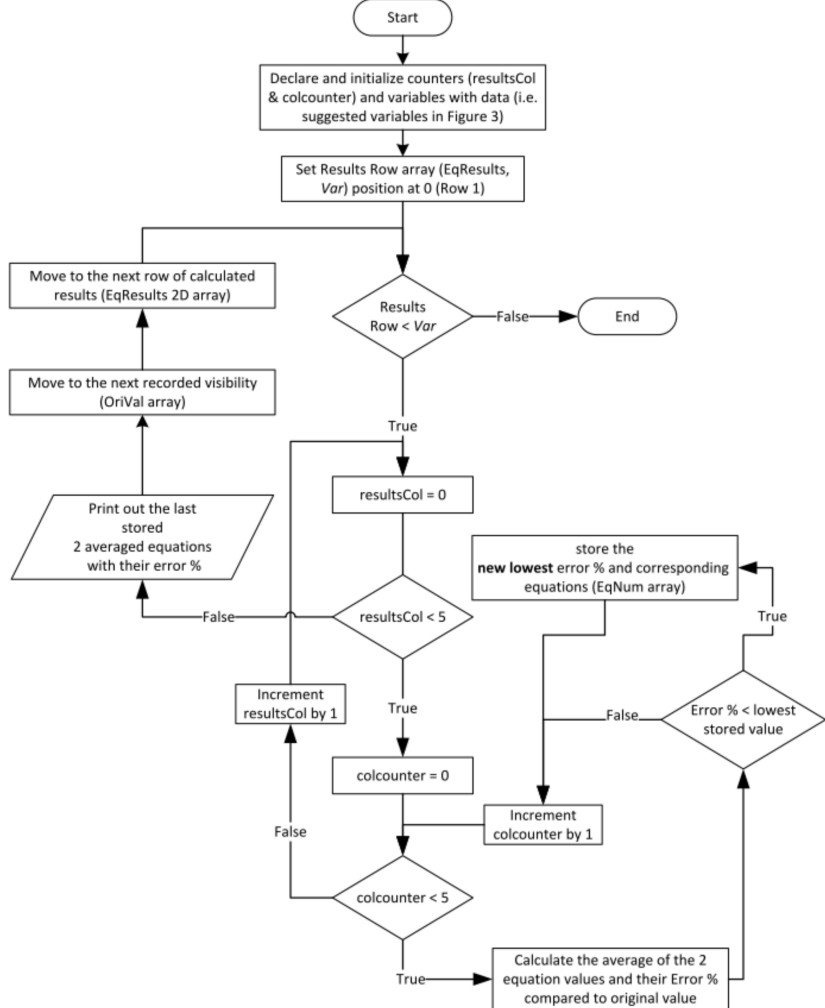

**Figure 4.** Flow chart of the designed computer program.

Each row is processed to locate the two most suitable (i.e., lowest percentage error with reference to an original value) averages of the summed empirical equations. Every empirical equation value is initially set as a constant and then summed and averaged with every other equation value in the same row. For example, the program starts with *Value 1-1* as a constant, then it is summed and averaged with each remaining value (*Value 1-2* to *Value 1-5*). Finally, the results are calculated for the percentage of error against the original value (*Value 1*) as follows:

$$Percentage\ of\ Error\ =\left|\frac{\frac{(value_{n-c})+(value_{n-v})}{2}}{Value\ n}-1\right|*100, \tag{18}$$

where $n$ indicates the row number to point to the values, $c$ is the number (1 to 5) to set the program to the corresponding value as a constant until it has been processed for percentage of error with the other values, and $v$ is a varied number (1 to 5) to point to values to be summed with the currently set constant.

## 3. Results and Discussions

A total of 143 measured TSP data points were retrieved and subsequently used to calculate visibilities using the equations in Table 1. In the first sub-section, the results are presented for general observations and analysis; these are followed by another sub-section highlighting the statistical significance of the results by suggesting optimistic scenarios and their probabilities.

### 3.1. General Observations and Analysis of the Results

The outcome of the program (C programming language; Visual Studio 2019) is defined under two terms; recurrences and averaged percentage of error, as seen in Figure 5. The recurrences refer to the number of combinations of two equations encountered that scored the lowest percentage of error, which is the second term, to that of the original. It is worth noting that Figure 5 demonstrates the best scores achieved in both cases for a single equation (i.e., E2, E3, E4, and E5) and the average of two summed equations (e.g., E3E5, E3E4, etc.). In addition, E1 is observed to have failed to achieve any significant results on its own as opposed to others.

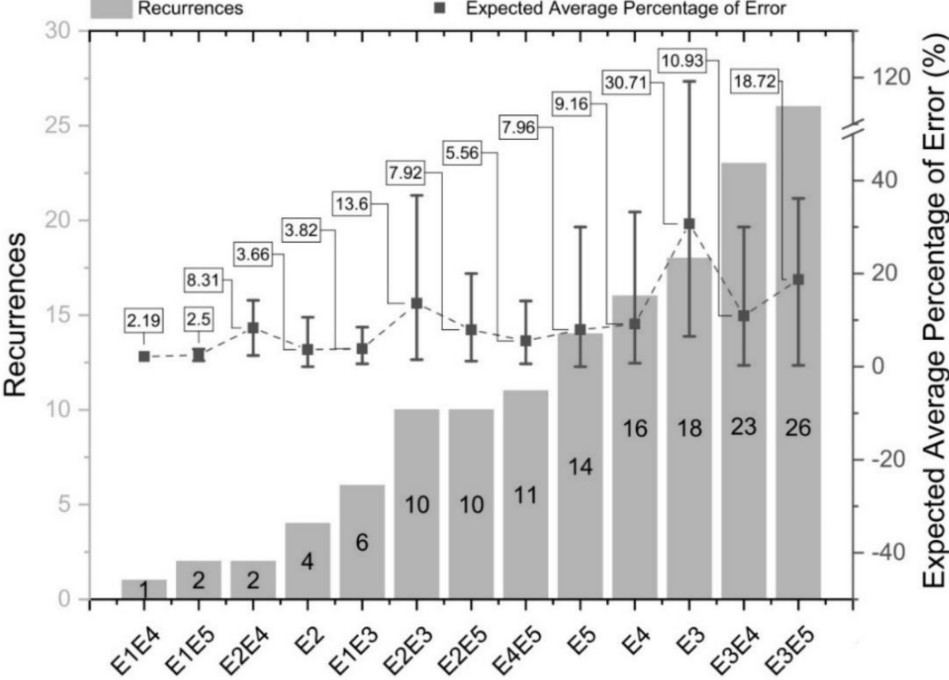

**Figure 5.** Illustrated outcome of equation recurrences scores based on the lowest expected percentage of error in estimating visibility at a given value of TSP concentration.

Sorting the results as described earlier revealed that averaging two summed equations may potentially lead to better estimations compared to single equations. The highest recorded recurrences (more than one-third of the results) were achieved by averaging E3E4 and E3E5, having relatively low average estimation errors of 10.93% and 18.72%, respectively.

Not far from expectations, the subsequent significant recurrences were scored by three single equations (i.e., E3, E4, and E5). The three single equations managed to score an accumulation of 48 recurrences with average estimation errors as low as 7.96% to 9.16% and as high as 30.71%.

Interestingly, from Figure 5, equations E3, E4, and E5, individually and in combination with each other, seem to be responsible for the most significant recurrences (75.5% of the results). In contrast, equations in combination with E1 and E2 were either marginally considerable or deemed negligible due to their meagre number of recurrences. This observation is particularly notable in three sources, as demonstrated in Figure 6, which is considered as a promising indicator. The reason being, the other three sources [3,14,15] were used to derive some of the empirical equations in Table 1, whereas sources

in Figure 6 are independent of any derivation process; hence, it is safe to assume the unbiased significance of these equations.

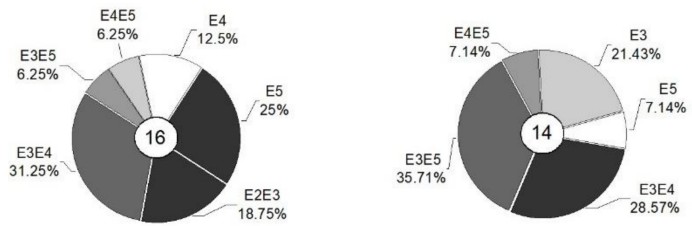

KSA dataset equation(s) recurrence % - Almuhanna [17,18]    USA dataset equation(s) recurrence % - Noll et al. [19]

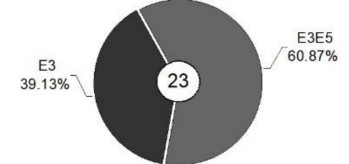

North-East Asia dataset equation(s) recurrence % - Chung et al. [20]

**Figure 6.** Observed dominance of E3, E4, and E5 equations, single or in-combination, compared to a complete absence of E1 and negligible contribution of E2. The numbers mediating the pie-charts are the number of data points retrieved from the source.

According to some studies, the relationship between visibility and dust concentration is influenced by a few factors such as dust particle size [24], extensivity of the data set used for derivations, measurement accuracy, and the distance of the measurement device from the source of the eroding soils [11].

While the geographical location of the experiment may account for some factors influencing empirical relationship development (e.g., distance from eroding soil), others are subject to the methods and equipment used. Previous studies exploring the relationship of visibility to TSP concentrations (e.g., Chepil and Woodruff's equation, E1; Patterson and Gillette's equation, E2) were based on field experiments. However, the relatively recent studies have become more innovative given the vast technological advancements for data collection, validation, and monitoring techniques (e.g., MODIS [25]).

For instance, Baddock et al.'s [15] (Equation E5) methodology employed data from a high-volume sampler and dust monitoring devices for improved concentration estimates and to derive instantaneous values of total suspended dust concentration from time-averaged values. Similarly, Shao et al.'s study [14], based on an extensive set of measurements, produced two expressions to accommodate two different dust-concentration-to-visibility relationship scenarios: one for distant dusts cases (threshold visibility ≥3.5 km) and the other for local dust (threshold visibility <3.5 km). This, in turn, may suggest that the combination of the prevailing equations (E3, E4, and E5) may have inherited such characteristics, thus resulting in estimations that are comparatively more accurate.

### 3.2. Statistical Significance

Despite the noted variations in Figure 5, it is apparent that averaging equations for visibility estimations exhibited noteworthy improvements, mainly if considered together with the resulting percentage of error. These conclusions are supported by the simple statistical analysis presented in Table 4, which describes the equation recurrences (average to maximum) with estimation percentage of errors (minimum to average) probabilities under two cases: in combination and singly.

**Table 4.** Statistical descriptions for equation recurrences in two states: singly and in combination, along with their estimation percentage of error probabilities.

| Set of Equations (Total Recurrences) | Statistical Description of Recurrences | | | | |
|---|---|---|---|---|---|
| | Mean (μ) | SD (σ) | Min. | Max. | P (μ < Z < Max.) |
| | Combination Equations | | | | |
| 9 (91) | 10.11 | 9.02 | 1 | 26 | 46.09% |
| | Single Equation | | | | |
| 4 (52) | 13 | 6.22 | 4 | 18 | 28.93% |
| | **Statistical Description of Estimation Error** | | | | |
| | Mean (μ), % | SD (σ), % | Min., % | Max., % | P (Min. < Z < μ) |
| | Average of Two Equations | | | | |
| 9 (91) | 11.66 | 9.99 | 0.28 | 36.8 | 37.27% |
| | Single Equation | | | | |
| 4 (52) | 15.87 | 19.20 | 0 | 119.18 | 29.58% |

The probabilities (*P*)—extracted based on the Z-score (Z) and normal distribution curve—of combination equations indicate that targeted range of recurrences (10.11 to 26) scored a probability of 46.09% with a chance of a relatively low estimation error (0.28% to 11.66%) rated at 37.27%. Conversely, single equations were found to have a probability of 28.93% for a range of 13 to 18 recurrences with a lesser chance (29.58%) of low estimation error.

For conformation, the combination equations in Figure 5 were evaluated using the sources in Table 2 and analyzed based on the technique used for Table 3, as shown in Table 5. In addition to substantiating E3, E4, and E5, the tabulated values indicate a considerable reduction in estimation error (reduced average RMSE) when in combination with each other.

**Table 5.** Performance (RMSE) of the algorithm-extracted combinations of equations.

| Source | E1E4 | E1E5 | E2E4 | E1E3 | E2E3 | E2E5 | E4E5 | E3E4 | E3E5 |
|---|---|---|---|---|---|---|---|---|---|
| KSA [17,18] | 19.93 | 18.66 | 13.29 | 15.91 | 9.31 | 12.04 | 3.02 | 1.42 | 2.09 |
| USA [3,19] | 84.77 | 93.21 | 68.30 | 77.88 | 61.59 | 76.93 | 13.68 | 11.66 | 8.69 |
| | 0.53 | 0.63 | 1.09 | 0.46 | 1.10 | 0.85 | 2.10 | 2.36 | 2.12 |
| North-East Asia [14,20] | 56.03 | 59.94 | 46.06 | 51.55 | 41.96 | 50.34 | 14.38 | 6.06 | 10.61 |
| | 30.72 | 30.81 | 22.15 | 25.97 | 17.85 | 22.48 | 6.30 | 6.04 | 7.17 |
| Australia [15] | 55.31 | 62.90 | 48.24 | 53.02 | 46.57 | 56.38 | 11.62 | 4.89 | 11.42 |
| Average | 41.22 | 44.36 | 33.19 | 37.46 | 29.73 | 36.50 | 8.52 | 5.40 | 7.02 |

Furthermore, despite the relatively poor performance of the other models, i.e., E1 and E2 in Table 3, pairing them with other equations may introduce a minor enhancement to their performance. Nonetheless, compared to E3, E4 and E5, they are deemed an unreliable estimator in our study.

In order to validate the importance of our findings, we can consider a case—provided in the next section—where a reported work of signal attenuation measurements in a dust storm lacked visibility information. Consequently, visibility estimation is needed with reasonable accuracy, thereby making the reported measurements benchmarkable for further investigations using any of the provided attenuation equations, i.e., Equations (12)–(17).

## 4. Employability Scenario: Assessing the Significance of the Enhanced Visibility Estimation Process

Mujlid [26], in her study on path losses of wireless sensor networks (WSNs) during sand/dust storms (SDS), measured signal attenuation under four environment conditions. These conditions were ambiguously labelled clear sky, dusty sky, sand storm, and heavy sand storm to indicate the severity of dust concentration at the experimental site.

However, the study set wind speed as an alternative meteorological representative of the environmental conditions, which may be used to derive particle concentrations (TSP). Therefore, before identifying visibilities, a preceding step is introduced to convert the meteorological condition into its particle-concentration equivalent, through which visibility calculation becomes possible.

### 4.1. Particle-Concentration Equivalent of a Dynamic Meteorological Condition (Wind Velocity)

As briefly stated earlier, Mujlid's [26] study demonstrated a correlation of path loss to the wind speed, which conversely can be understood as a link to the severity of the SDS. The reported wind speeds were 0.6 m/s, 3.6 m/s, 3.8 m/s, and 7.3 m/s, respectively, for the environmental conditions of clear sky, dusty sky, sand storm, and heavy sand storm. Although the specifications of the instrument used for wind speed measurements are not reported in [26], it is believed that it is of a similar make (JL-FS2 anemometer) to the one reported in this source [27] with an error of ±3%. Therefore, the recorded speeds are expected to have reading errors of ±0.018 m/s, ±0.108 m/s, ±0.114 m/s, and ±0.219 m/s, respectively.

Since wind is the dynamic condition of dust storms [28], and dust storms are fundamentally a concentration of particles in the air, it is plausible to establish a relationship between the number of particle concentrations and corresponding wind speeds, as has been demonstrated by the authors of [29] using the following expression:

$$M = 29.66e^{0.7u} \ \mu\text{g}/\text{m}^3, \tag{19}$$

where $M$ is the dust concentration in $\mu\text{g}/\text{m}^3$ and $u$ is the wind speed in m/s. By substituting the reported wind speeds (and estimated errors) in the order mentioned in the earlier paragraphs into Equation (13), we get the following values: 45.14 (±0.57) $\mu\text{g}/\text{m}^3$, 316.2 (−26.84, +28.95) $\mu\text{g}/\text{m}^3$, 424.03 (−32.52, +35.22) $\mu\text{g}/\text{m}^3$, and 4913.78 (−698.38, +814.09) $\mu\text{g}/\text{m}^3$, respectively.

Equation (19) provided by K. Kandler et al. [29] was superficially mentioned without elaborating on its employability restrictions. However, assuming the same study environment as in [29], a recently published work [30] indicated that variation in SDS particle concentrations are not only limited to changing wind speeds but can be observed in their vertical distribution as well. Wei et al.'s [30] real-time measurements revealed that during the SDS period for heights less than 100 m, the concentrations ranged between 1220 $\mu\text{g}/\text{m}^3$ and 42,146 $\mu\text{g}/\text{m}^3$. Meanwhile, the lowest measured concentration in the same event at its initial meteorological conditions leading up to the storm was an average of 31.62 $\mu\text{g}/\text{m}^3$ at heights less than 200 m.

Wei et al.'s [30] results confer greater credibility on the results obtained from Equation (13). Therefore, for evaluation purposes, the values may be adopted conveniently, considering that the observation height for Mujlid's [26] environment conditions could not have exceeded 1.7 m, which is the average human height [31].

### 4.2. Evaluating the Proposed Visibility Calculation Method

With the dust concentrations ($M$) obtained from Equation (19), visibilities ($V$) can be calculated using any combination of visibility equations adhering to our findings. Having the lowest average RMSE, the averaged sum of E3 and E5 is taken as an example for evaluation, and the results are presented in Table 6 in addition to the calculated dust concentrations and wind speeds with corresponding inaccuracies.

**Table 6.** Corresponding dust concentrations and visibilities for the reported wind speeds in [26].

| Wind Speed, m/s | Calculated Dust Concentration, $\mu\text{g}/\text{m}^3$ | Calculated Visibility, km |
| --- | --- | --- |
| 0.6 (±3%) | 45.14 (±0.57) | 30.51 (−0.24, +0.25) |
| 3.6 (±3%) | 316.2 (−26.84, +28.95) | 9.9 (−0.53, +0.55) |
| 3.8 (±3%) | 424.03 (−32.52, +35.22) | 8.16 (−0.45, +0.46) |
| 7.3 (±3%) | 4913.78 (−698.38, +814.09) | 0.43 (−0.046, +0.087) |

In addition to scattered evidence [32], the visibilities in Table 6 and the environmental conditions in [26] can be further supported by considering a meteorological standard that was set for reporting purposes. In 1979, the China Meteorological Administration (CMA) [33–35] established a standard

for dividing SDS outbreak sequences into four levels. Each level is a description of a meteorological process associated with a projected range of horizontal visibility, and they are as follows: (1) floating dust ($V < 10$ km); (2) blowing dust (1 km $\leq V \leq 10$ km); (3) sand/dust storm (0.5 km $\leq V \leq 1$ km); and (4) severe sand/dust storm ($V < 0.5$ km).

In comparison, of the four environment conditions defined by [26], three conditions, along with the calculated visibilities, despite the associated errors, exhibit a high degree of range-compliance in accord with the CMA's levels (Table 7). It is worth noting that the clear sky condition was not included since it corresponds to unobstructed horizontal visibility that extends beyond 10 km.

**Table 7.** Environment conditions labelled by [26] with the corresponding calculated visibilities and their similarities to the China Meteorological Administration's standards.

| Environment Classification and Wind Speed Measurements by [26] | | | Standards Defined by the China Meteorological Administration (CMA) | |
|---|---|---|---|---|
| Environment condition | Wind speed, m/s | Calculated visibility, km | Environment condition | Expected horizontal visibility, km |
| Dusty sky | 3.6 (±3%) | 9.9 (−0.53, +0.55) | Floating dust | <10 |
| Sand storm | 3.8 (±3%) | 8.16 (−0.45, +0.46) | Blowing sand | 1–10 |
| Heavy sand storm | 7.3 (±3%) | 0.43 (-0.046, +0.087) | Dust/sand storm | 0.5–1 |
| | | | Severe dust/sand storm | <0.5 |

Furthermore, it has been statistically inferred that for a span of two decades, the lowest storm-erupting wind speed was 5 m/s [36], which validates the 4th tabulated environment condition since the reported wind velocity was 7.3 m/s. Besides, with visibility of 0.56 km, it shows exceptional conformity to the CMA's specified visibility projections ($0 \leq$ visibility $\leq 1$ km) for SDS events.

As for the remaining conditions with wind velocities below 5 m/s, it is still considered within an acceptable CMA classification range given the overlapping projections of horizontal visibility ranges, which are generally below 10 km.

## 5. Conclusions

This paper aimed to address the claims of visibility over- and under-estimation tendencies resulting from existing empirical equations. To that end, an algorithm was developed to assist in processing various data sets of TSPs and recorded visibilities, and output the best fitting results in terms of equation recurrences accompanied by their estimation percentage of error. The sources consisted of variations in terms of data collection techniques, sites of interest (urban, rural, or regional level) with particulate concentrations as low 25 ug/m$^3$ and as high as 440 mg/m$^3$ and corresponding visibilities as high as 56 km and as low as 0.06 km.

While the results did not yield a decisive conclusion, they did partially substantiate the Tews [13] (average RMSE of 7.16), Shao et al. [14] (average RMSE of 7.07), and Baddock et al. [15] (average RMSE of 11.7) equations, which emerged as reliable estimators of visibility with reasonable accuracy compared to other empirical models (average RMSE > 29). Furthermore, by using our technique, we found that by averaging Tews [13] and Shao et al. [14], Tews [13] and Baddock et al. [15], and Shao et al. [14] and Baddock et al.'s [15] equations, average RMSE could be further reduced to 5.4, 7.02, and 8.52, respectively.

Finally, to evaluate the proposed approach, we provided a demonstration of employability, where a study provided signal attenuation measurements at an experimental site with vaguely defined meteorological conditions. Consequently, the conditions were converted into visibility using our findings and found to be in a good agreement with projected ranges of horizontal visibility standards set by the China Meteorological Administration (CMA).

**Author Contributions:** Conceptualization, H.N.M. and W.I.; methodology, H.N.M.; software, H.N.M.; validation, H.N.M., and W.I.; formal analysis, H.N.M.; investigation, H.N.M.; resources, H.N.M.; data curation, H.N.M.; writing—original draft preparation, H.N.M.; writing—review and editing, W.I.; supervision, W.I.; project administration, W.I.; funding acquisition, W.I. All authors have read and agreed to the published version of the manuscript.

**Funding:** This research was funded by Universiti Sains Malaysia Research University Individual (RUI) Grant, grant number 1001/PELECT/8014058.

**Acknowledgments:** Special appreciation to the Malaysia Ministry of Higher Education for the support and School of Electrical and Electronics Engineering, Engineering Campus, USM for the assistance and space in conducting the research.

**Conflicts of Interest:** The authors declare no conflict of interest.

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
