# Peer review of "Visibility Parameter in Sand/Dust Storms’ Radio Wave Attenuation Equations: An Approach for Reliable Visibility Estimation Based on Existing Empirical Equations to Minimize Potential Biases in Calculations"

_applsci, doi:10.3390/app10217530_

Round 1
Reviewer 1 Report
This work acts as a small review regarding visibility degradation due to ambient aerosol. The authors evaluate 5 equations that relate visibility to aerosol mass concentration and evaluate their performance against measurements. Taking it one-step further, the authors suggest a combination of these equations that model optimally visibility. This work is well structured, concise and informative and should be published after several changes have been applied.
In most cases when a model, empirical equations in this case, are compared against measurements to review the performance of the model, a set of metrics such as the z-score, the agreement index, RMSE or other are employed. The authors have chosen a rather unique method of assessing their results and it is worthwhile to state why they decided to use this metric and provide a reference for it.
The authors either neglected, or ignore, that visibility is a very well defined quantity that related to aerosol scattering. For a visual contrast of 0.02 (some authors use a higher value) visibility is equal to V=3.912/bext where bext is the extinction coefficient.
The extinction coefficient is a function of many parameters, which the authors mention in their work, such as the refractive index and hence composition along with particle size and number concentration related to that size. Please read Seinfeld and Pandis, 2016 Chapter 15. This should be mentioned somewhere in the introduction. Also please reformulate lines 191-192 accordingly.
Because scattering measurements are relatively scarce several studies tried to link bulk aerosol mass concentration, without any information on size, refractive index or composition to visibility. In many cases, these equations apply to aerosol that hold specific features with respect to the above physical quantities. The authors briefly acknowledge that in lines 205-207. It is therefore important to mention in table 1 not only the study but also hint the reader as to what type of aerosol this study was based on. The authors mention dust particles frequently in the manuscript but should keep in mind that haze is a major factor for visibility degradation. It is also important to mention if the empirical equations were based on ambient or laboratory data. Finally for ambient studies it may be worthwhile to mention location.
The same job should be done for Table 2 so that is there is a direct comparison. I am heavily under the impression that this information will explain why one model performs better than the other. It will also explain why the TSP values are so high in Figure 1.
Having in mind Figure 1 the authors should state the TSP concentration range their model evaluation has been based on. It seems that this study has focused on high loadings exceeding 50 μg/m3. Please discuss.
Section 4 is very problematic. The authors report agreement between an even more ambiguous empirical equation (13) and the observation standards by the China Meteorological society. This is fine as long as it is clearly stated that this agreement and empirical equation only holds under certain assumptions. These assumptions must be clear. The authors do mention that these are valid only for SDS but they must also clarify the conditions these equations are applicable. Currently the manuscript is formulated in a way suggesting that Eq 13 and 14 are always valid.
Table 3 is very confusing. Why the set of equations are 9 since only 5 are mentioned in Table 1?
H. Seinfeld, and S. N. Pandis, Atmospheric Chemistry and Physics: From Air Pollution to Climate Change, John Wiley and Sons, 2016
Author Response
We would like to thank the reviewers for their comments, suggestions and efforts towards improving this manuscript. Our reply to the comments are as follows:
|
Reviewer 1 (in bold; reviewer’s comments/suggestions, in non-bold; authors’ responses) |
“In most cases when a model, empirical equations in this case, are compared against measurements to review the performance of the model, a set of metrics such as the z-score, the agreement index, RMSE or other are employed. The authors have chosen a rather unique method of assessing their results and it is worthwhile to state why they decided to use this metric and provide a reference for it.”
We understand the reviewer’s concern; however, the use of such method was based on basic statistics (mean, standard deviation, min, max and probability) which stems from the direct interest in calculating the probability of recurrences with respect to its associated error scores. However, an additional step has been taken - as implicitly suggested by the reviewer – to provide the Root-Mean-Square of error (RMSE); Table 3 and Table 5.
“The authors either neglected, or ignore, that visibility is a very well-defined quantity that related to aerosol scattering. For a visual contrast of 0.02 (some authors use a higher value) visibility is equal to V=3.912/bext, where bext is the extinction coefficient. The extinction coefficient is a function of many parameters, which the authors mention in their work, such as the refractive index and hence composition along with particle size and number concentration related to that size. Please read Seinfeld and Pandis, 2016 Chapter 15. This should be mentioned somewhere in the introduction. Also please reformulate lines 191-192 accordingly.”
We thank the reviewer for taking the time to share some insights from the realm of optical visibility and the basic mathematics governing it. Unfortunately, given the short amount of time to conduct the necessary revisions, we were not able to get a hold of the mentioned book.
however, attempting to meet the reviewer’s expectation, the following additions have been made:
- Line 57-58: the 15 dB proportionality has been explained and related to its origin, which the measured median (i.e. 0.031) of normalized difference luminances between a mark and a reference background.
- Line 63-70: explaining the derivation process of Equation (5), which, indirectly, leads to the reviewer’s comment on what is defined as Koschmieder visibility law.
“It is therefore important to mention in table 1 not only the study but also hint the reader as to what type of aerosol this study was based on.”
Table 1 has been partially reworked to include the suggested additions. The additions include data collection technique, monitoring period, location, number of field observations and distance from eroding soils.
“The authors mention dust particles frequently in the manuscript but should keep in mind that haze is a major factor for visibility degradation. It is also important to mention if the empirical equations were based on ambient or laboratory data. Finally, for ambient studies it may be worthwhile to mention location.”
In response to the reviewer’s concern, we have elaborated on the employed empirical equations under Table 1 to define the condition under which it was developed.
“The same job should be done for Table 2 so that is there is a direct comparison. I am heavily under the impression that this information will explain why one model performs better than the other. It will also explain why the TSP values are so high in Figure 1.”
Table 2 has been partially reworked and expanded to accommodate the reviewer’s suggestions.
“Having in mind Figure 1 the authors should state the TSP concentration range their model evaluation has been based on. It seems that this study has focused on high loadings exceeding 50 μg/m3. Please discuss.”
There were no intentions of directing our work towards specific amounts of loading. In fact, the interest was mainly in variations among the datasets themselves to probe for how accommodating the model’s range are for scenarios such as different visibilities corresponding to somewhat similar TSP quantities and vice versa. However, what probably caused an unintentional choice of high loadings is that the current study centres around the sandy/dusty environment.
Further, we would like to draw the reviewer’s attention to the following: After reviewing our data, we realised duplicates were included by mistake (by default, only 16 data points should be considered in Figure 2f). To that end, we would like to declare the following changes, which had minor effects on the outcome:
- The data has been reprocessed and replotted in Figure 5 (same outcome, with minor offsets). The figure also is fitted with error bars for clarity and ease of interpretations.
- Values were updated in (line 207,209-210,212, Figure 5, Figure 6, Table 1, Table 4, line 252-255, line 22,24).
- In figure 2f, the left-most data points were too close to each other, implying a count of two points only, while there are three. Thus, as a workaround, the three data points, while maintaining the colour to match the provided legend, were varied in size and opacity.
- The current total data points are 143.
Please accept our sincere apologies, and we very much appreciate the efforts in highlighting this part.
“Section 4 is very problematic. The authors report agreement between an even more ambiguous empirical equation (13) and the observation standards by the China Meteorological society. This is fine as long as it is clearly stated that this agreement and empirical equation only holds under certain assumptions. These assumptions must be clear. The authors do mention that these are valid only for SDS but they must also clarify the conditions these equations are applicable. Currently the manuscript is formulated in a way suggesting that Eq 13 and 14 are always valid.”
Equation 14 has been removed and replaced with in-text mention of the employed equation based on the lowest average RMSE (Line 310-311).
Furthermore, Equation 14 (now 19; line 291) has been commented on explicitly declared for evaluation purposes only (line 296).
A reference ([32]) has been added (line 319) to indicate that China Meteorological Administration is only taken as a guide line for general SDS levels classification.
“Table 3 is very confusing. Why the set of equations are 9 since only 5 are mentioned in Table 1?”
Set of equations refer to Figure 5.
e.g. 9(91) refers to 9 in-combination equations with a total of 91 recurrences.
Reviewer 2 Report
Review of the manuscript entitled “Visibility Parameter in Sand/Duststorms’ Radio Wave Attenuation Equations: An Approach for Reliable Visibility Estimation to Minimize Potential Biases in Calculations”.
The manuscript describes a reasonable approach to estimate the visibility parameter during sand/dust storms by using an average of two empirical expressions. The pair of empirical formulas is found by applying the algorithm developed in this work. This approach improves the finding of correct results by comparing calculated values with recorded visibilities. The manuscript is well organized and clearly written with some exceptions (minor comments).
However, the error analysis is not as developed as I would expect. Measured quantities are written without any uncertainties (Tab.2, Tab.4, Fig.2). The similar conclusions to those presented in the manuscript could be inferred if the errors of recorded visibilities are plotted in Fig.2. It will be clear what calculated visibilities obtained from empirical formulas are within in the error bar of recorded visibilities. And finally which empirical formulas dominate to estimate better the recorded visibilities. It could be another confirmation of the algorithm applied in this manuscript to find results closest to the recorded ones. The errors of calculated visibilities along with errors of dust mass concentrations should be also shown in Fig.2. But I am not sure if for clarity of Fig.2 it should be presented. Maybe it is enough to write in the text or in a table.
I summarized my main comments below.
- Different models from (6) to (11), presented in the manuscript, do not have any evident connection with empirical formulas in Tab.1. I think it should be shown how these models relate to empirical formulas in Tab.1 through equation (5).
- There is no paragraph with an analysis of error for results given in Tab.4. I suggest to calculate uncertainties from the error propagation formula by assuming the uncertainty of measured wind speed. The error of wind speed is not written in the quoted papers [23] and [25] although it should be. Maybe this value (error) is provided in the other papers? If not, it is better to assume some reasonable uncertainty or accuracy of wind speed. The similar analysis of error should also be performed for Tab.1 using errors of measurements in Tab.2. Eventually, the discussion of uncertainties should appear in the text.
Minor comments:
Line 55: There is no unit for dust mass concentration M. Please, provide it. Moreover, visibility V in formula (5) is the same like in expressions from (6) to (11). Please, write subscript ‘b’ in formula (5) or delete the subscript ‘b’ to be consistent with equations written further.
Line 56: The unit for constant C is wrong. According to the paper by Patterson and Gillette (1977) the constant C has unit [g m-3 km].
Line 57: The sentence starting from “This relationship…” is not clear. Which relationship, (5), (4) or (2)? Could you show it?
Line 78: You introduced a new name for the same quantity which appeared in equation (5) - dust mass concentration (M). Maybe, to be consistent with equation (5) it is better to write everywhere D or M or TSP because now it is misleading.
Line 103: Table 2 contains measured variables. Please, provide estimated error of these measurements.
Line 126: I think it is better to plot also errors (uncertainties) for recorded visibilities to see how far plotted equations given in Tab.1 are from recorded visibilities in Fig.2.
In Fig.2a you plotted 46 points but 47 points are considered in Tab.1. What happened with one point? The similar problem is found in Fig.2f. There are only 15 points instead of 24 points given in Tab.1. Why did not you include all points in Fig.2f? Fig.2e has one odd point for Shao et al. formula for visibility larger than 3.5 km. Why is it plotted?
Line 169: It will be clearer if you write why E1 equation is not used in the analysis and in Fig.5 in the text.
Line 181: The sentence “Interestingly…” is not clear. Equations E3, E4 and E5 individually or in combination are not responsible for 76.8% of the results.
Line 182: The sentence should refer to Fig.5.
Line 184: The second part of the sentence is not clear. What is your conclusion? What is the indicator of what, which is promising? What do you want to say in this sentence?
Line 233: It should be written ‘sets’ and ‘meteorological’.
Line 249: ‘vertical’ sounds better than altitudinal.
Line 292: It is better to write what the reasonable accuracy is like?
Author Response
We would like to thank the reviewers for their comments, suggestions and efforts towards improving this manuscript. Our reply to the comments are as follows:
|
Reviewer 2 (in bold; reviewer’s comments/suggestions, in non-bold; authors’ responses) |
“Which empirical formulas dominate to estimate better the recorded visibilities. It could be another confirmation of the algorithm applied in this manuscript to find results closest to the recorded ones.”
Two tables were provided highlighting the performance level among various models in terms of RMSE; one table for single equations (Table 3) and in-combination (table 5; based on figure 5), based on which a recommendation has been made and was rectified in the conclusion section.
“The errors of calculated visibilities along with errors of dust mass concentrations should be also shown in Fig.2. But I am not sure if for clarity of Fig.2 it should be presented. Maybe it is enough to write in the text or in a table.”
A table of RMSE analysis is provided (Table 3).
“Different models from (6) to (11), presented in the manuscript, do not have any evident connection with empirical formulas in Tab.1. I think it should be shown how these models relate to empirical formulas in Tab.1 through equation (5).”
More hints on visibility added (line 57-58).
A reworked segment linking equation 5 to the formulation of an attenuation expression (Equation (11)) (line 63-80).
“There is no paragraph with an analysis of error for results given in Tab.4. I suggest to calculate uncertainties from the error propagation formula by assuming the uncertainty of measured wind speed. The error of wind speed is not written in the quoted papers [23] and [25] although it should be. Maybe this value (error) is provided in the other papers? If not, it is better to assume some reasonable uncertainty or accuracy of wind speed. The similar analysis of error should also be performed for Tab.1 using errors of measurements in Tab.2. Eventually, the discussion of uncertainties should appear in the text.”
The error of the anemometer was introduced into the text (line284-287). This, in turn, has been taken into consideration, and the visibilities have been calculated accordingly (Table 6 (previously 4)).
“Line 55: There is no unit for dust mass concentration M. Please, provide it. Moreover, visibility V in formula (5) is the same like in expressions from (6) to (11). Please, write subscript ‘b’ in formula (5) or delete the subscript ‘b’ to be consistent with equations written further.”
The unit for M has been added, and subscript “b” has been discarded from all equations.
“Line 56: The unit for constant C is wrong. According to the paper by Patterson and Gillette (1977) the constant C has unit [g m-3 km].”
The unit has been corrected.
“Line 57: The sentence starting from “This relationship…” is not clear. Which relationship, (5), (4) or (2)? Could you show it?”
A reworked segment linking equation 5 to the formulation of an attenuation expression (Equation (11)) (line 63-80).
“Line 78: You introduced a new name for the same quantity which appeared in equation (5) - dust mass concentration (M). Maybe, to be consistent with equation (5) it is better to write everywhere D or M or TSP because now it is misleading.”
The letter signifying mass concentration has been unified to M, in addition to hinting an occasional use of TSP as an alternative (line 100).
“Line 103: Table 2 contains measured variables. Please, provide estimated error of these measurements.”
Table 2 has been expanded to include reported measurement errors (where possible).
“Line 126: I think it is better to plot also errors (uncertainties) for recorded visibilities to see how far plotted equations given in Tab.1 are from recorded visibilities in Fig.2.”
An attempt has been carried out to execute the suggestion. Unfortunately, a ±10% of inaccuracy (for example, as reported in table 2) barely noticeable on a logarithmic scale. Alternatively, we would like to cite the reviewer’s comment made earlier to include a table of errors – In this case, it is RMSE analysis found in table 3.
“In Fig.2a you plotted 46 points but 47 points are considered in Tab.1. What happened with one point? The similar problem is found in Fig.2f. There are only 15 points instead of 24 points given in Tab.1. Why did not you include all points in Fig.2f? Fig.2e has one odd point for Shao et al. formula for visibility larger than 3.5 km. Why is it plotted?”
For Fig.2a, 2e: the missing point has been added/data point linked on the plot.
For Fig.2f, we genuinely thank the reviewer for this level of plot inspection. After reviewing our data, we realised duplicates were included by mistake (by default, only 16 data points should be considered). To that end, we would like to declare the following changes, which had minor effects on the outcome:
- The data has been reprocessed and replotted in Figure 5 (same outcome, with minor offsets). The figure also is fitted with error bars for clarity and ease of interpretations.
- Values were updated in (line 207,209-210,212, Figure 5, Figure 6, Table 1, Table 4, line 252-255, line 22,24).
- In figure 2f, the left-most data points were too close to each other, implying a count of two points only, while there are three. Thus, as a workaround, the three data points, while maintaining the colour to match the provided legend, were varied in size and opacity.
- The current total data points are 143.
Please accept our sincere apologies.
“Line 169: It will be clearer if you write why E1 equation is not used in the analysis and in Fig.5 in the text.”
A note has been added (line 199-200).
“Line 181: The sentence “Interestingly…” is not clear. Equations E3, E4 and E5 individually or in combination are not responsible for 76.8% of the results.”
Sentence corrected (line 211-212).
“Line 182: The sentence should refer to Fig.5.”
Sentence referring to figure 5 has been added (line 211).
“Line 184: The second part of the sentence is not clear. What is your conclusion? What is the indicator of what, which is promising? What do you want to say in this sentence?”
The sentence has been rephrased (line 214-218).
“Line 233: It should be written ‘sets’ and ‘meteorological’.”
The words have been corrected.
“Line 249: ‘vertical’ sounds better than altitudinal.”
The word has been changed.
“Line 292: It is better to write what the reasonable accuracy is like?”
RMSE values are added into conclusion (Line 348-354).
